# Loss of Human Beta Cell Identity in a Reconstructed Omental Stromal Cell Environment

**DOI:** 10.3390/cells11060924

**Published:** 2022-03-08

**Authors:** Blandine Secco, Kevin Saitoski, Karima Drareni, Antoine Soprani, Severine Pechberty, Latif Rachdi, Nicolas Venteclef, Raphaël Scharfmann

**Affiliations:** 1Institut Cochin, Université de Paris, INSERM U1016, CNRS UMR 8104, 75014 Paris, France; blandine130389@hotmail.fr (B.S.); kevin.saitoski@inserm.fr (K.S.); severine.pechberty@inserm.fr (S.P.); latif.rachdi@inserm.fr (L.R.); 2Cordeliers Research Centre, INSERM, Immunity and Metabolism in Diabetes Laboratory, Université de Paris, 75006 Paris, France; karimadrareni@gmail.com (K.D.); antoinesoprani@hotmail.com (A.S.); nicolas.venteclef@upmc.fr (N.V.); 3Clinique Geoffroy Saint-Hilaire, Ramsey General de Santé, 75005 Paris, France

**Keywords:** EndoC-βH1, dedifferentiation, stromal cells, type 2 diabetes, adipose tissue, human pancreatic beta cells

## Abstract

In human type 2 diabetes, adipose tissue plays an important role in disturbing glucose homeostasis by secreting factors that affect the function of cells and tissues throughout the body, including insulin-producing pancreatic beta cells. We aimed here at studying the paracrine effect of stromal cells isolated from subcutaneous and omental adipose tissue on human beta cells. We developed an in vitro model wherein the functional human beta cell line EndoC-βH1 was treated with conditioned media from human adipose tissues. By using RNA-sequencing and western blotting, we determined that a conditioned medium derived from omental stromal cells stimulates several pathways, such as STAT, SMAD and RELA, in EndoC-βH1 cells. We also observed that upon treatment, the expression of beta cell markers decreased while dedifferentiation markers increased. Loss-of-function experiments that efficiently blocked specific signaling pathways did not reverse dedifferentiation, suggesting the implication of more than one pathway in this regulatory process. Taken together, we demonstrate that soluble factors derived from stromal cells isolated from human omental adipose tissue signal human beta cells and modulate their identity.

## 1. Introduction

The endocrine pancreas plays a crucial role in nutritional homeostasis through the synthesis and secretion of hormones by cells aggregated into islets of Langerhans. The islets of Langerhans are endocrine micro-organs implicated in glycemic regulation. They are dispersed in the pancreatic gland. Each islet (1000 in a mouse pancreas, 1 million in a human pancreas) contains five different cell subtypes: alpha-, beta-, delta-, PP- and epsilon-cells, which produce and secrete glucagon (GCG), insulin (INS), somatostatin (SST), pancreatic polypeptide (PPY) and ghrelin (GHRL), respectively. Diabetes is characterized by high blood glucose levels, which, in most cases, result from the inability of the pancreas to secrete sufficient amounts of insulin. Type 1 diabetes (T1D) is caused by the autoimmune-mediated destruction of insulin-producing beta-cells [1]. Type 2 diabetes (T2D) is a life-threatening metabolic disease currently attaining an epidemic scale. T2D develops when insulin-producing pancreatic beta cells fail to respond to the increasing insulin demand produced by peripheral insulin resistance in skeletal muscle and adipose tissue [2]. Although genetic and environmental factors contribute to the risk of developing T2D [3], the molecular basis of the disease is incompletely understood. More specifically, the fate of beta cells in T2D remains a matter of intense discussion [4]. While data are available that indicate that the insulin-positive cell mass is lower in the pancreas of deceased T2D people when compared to controls [5,6,7], the impact of decreased beta cell mass in type 2 diabetes remains debated [8], and more work needs to be done on that topic. While it has been proposed that the beta cell mass decreases in T2D patients due to premature programmed cell death [9,10], recent data challenge the beta cell death hypothesis. They suggest that beta cell dedifferentiation represents an alternative mechanism to explain the insufficient insulin production observed in type 2 diabetes [11,12]. This concept is now supported by experiments performed in a number of experimental models [13,14,15]. However, information on human beta cell dedifferentiation remains scarce, when compared to that obtained in mouse models [16,17]. We previously developed functional human beta cells lines [18,19] and used them to model human beta cell dedifferentiation under pathophysiological conditions [20,21]. 

In a number of pathophysiological contexts, adipose tissue has an important function in disturbing glucose homeostasis by secreting factors that affect the function of cells and tissues throughout the body, including beta cells [22,23]. In obesity, adipose tissue undergoes morphological and cellular changes, leading to adipocyte hypertrophy and adipose tissue inflammation, which represent hallmarks of maladaptive adipose tissue expansion in obesity [24]. This pathological expansion of adipose tissue is more often observed in visceral adipose tissue than in subcutaneous adipose tissue. Interestingly, adipose stromal cells from obese adipose tissue acquire an atypical (or inflamed) phenotype that propagates pathogenic signals to other organs, including the pancreas [25,26,27,28]. Indeed, pancreatic fat has been proposed to regulate insulin production and beta cell function [29,30]. However, the pathophysiological impacts and the cellular mechanisms by which adipose tissue impacts human beta cell fate are poorly understood [31]. In the present study, we have focused on the interactions between adipose tissue and human beta cells. We have reasoned that secreted signals derived from human adipose tissue might affect human beta cells. We demonstrate here that soluble factors derived from stromal cells isolated from human adipose tissue signal human beta cells through different pathways, such as STAT, SMAD and NFKB, previously shown to be involved in beta cell function [21,32,33,34,35]. We also observed that such soluble factors decrease the expression of beta cell-specific transcription factors, such as MAFA, PDX1 and NKX6-1 [36], while inducing at the same time the expression of progenitor markers such as SOX9 [37], suggesting a beta cell dedifferentiation process [20].

## 2. Materials and Methods

### 2.1. Culture of EndoC-βH1 Cells

Human pancreatic beta cell line EndoC-βH1 cells were cultured in low-glucose DMEM medium (5.6 mmol/L) (Thermo Fisher Scientific, Carlsbad, USA) supplemented with 2% BSA fraction V (Roche Diagnostics, Basel, Switzerland), 50 μM β-mercaptoethanol (Sigma-Aldrich, St. Louis, MO, USA), 10 mM nicotinamide (Calbiochem for Merck, Darmstadt, Germany), 5.5 μg/mL human transferrin (Sigma-Aldrich), 6.7 ng/mL selenite (Sigma-Aldrich), 100 U/mL penicillin and 100 μg/mL streptomycin (Thermo Fisher Scientific). Cells were seeded at 8.10^4^ cells/cm^2^ on plates coated with 1.2% Matrigel (Sigma-Aldrich) and 3 µg/mL fibronectin (Sigma-Aldrich). The cells were cultured at 37 °C and 5% CO_2_, and passaged once a week.

### 2.2. Isolation, Culture of Adipose-Derived Stromal Cells and Production of Conditioned Media (CM)

Fragments of subcutaneous and omental adipose tissue were obtained from obese patients undergoing bariatric surgery at Geoffroy Saint-Hilaire hospitals (n = 26, age = 40.8 ± 12.1 years, BMI: 43.1 ± 7.2 kg/m^2^). The clinical characteristics of patients are provided in Appendix A. The study was conducted in accordance with the Helsinki Declaration. The Ethics Committee of CPP Ile-de-France approved the clinical investigations for all individuals, and written informed consent was obtained from all individuals. 

Human adipose tissue fragments were minced and digested with collagenase type II (Sigma aldrich) for 1 h at 37 °C, centrifuged, incubated with red blood cells lysis buffer and filtered through a 70 µM filter before being plated. The cells were amplified in a medium composed of 50% Human Preadipocyte Growth Medium (Sigma Aldrich) and 50% high glucose (25 mmol/L) DMEM medium (Thermofisher Scientific) with 10% fetal bovine serum (EUROBIO, Courtaboeuf, France) for 4–6 days before being plated and used for experiments. Conditioned media were prepared from cells, and were cultivated until confluence in either the absence or presence of TNFα (1000 U/mL, R&D). Conditioned media were also prepared from adipose-derived stromal cells transformed into adipocytes. Then, confluent adipose-derived stromal cells were cultured for 4–5 days in an adipogenic cocktail medium composed of 50% Human Adipocyte Differentiation Medium (Sigma Aldrich) and 50% high-glucose (25 mmol/L) DMEM medium, 10% fetal bovine serum, insulin (830 nM, Sigma Aldrich), IBMX (0.5 µM, sigma Aldrich), Rosiglitazone (5 µM, Sigma Aldrich), and dexamethazone (1 µM, sigma Aldrich). The cells were then maintained for 2–3 days in the same medium lacking IBMX, rosiglitazone and dexamethazone. Once the cells reached the desired stage, they were washed twice with PBS and cultivated in EndoC-βH1 culture medium for 48 h for CM production, which were filtered and frozen. CM were used on EndoC-βH1 cells diluted by half in the absence or presence of TNFα (1.000 U/mL, R&D). The CM’s effect on EndoC-βH1 cells survival was measured by plating 3.5 × 10^5^ cells/well. After 48 h of treatment with CM diluted by half in the absence or presence of TNFα (1.000 U/mL, R&D), cell numbers were measured with an Invitrogen™ Countess™ 3 FL Automated Cell Counter (Thermofisher Scientific).

### 2.3. SiRNA Transfection

EndoC-βH1 cells were transfected as described [21] in OptiMEM using Lipofectamine RNAiMAX (Life Technologies, Carlsbad, USA) with siRNA SMARTpools (Horizon Discovery LTD, Cambridge, UK). The medium was replaced 2.5 h later with fresh culture medium and analyses were carried out 4 days post-transfection. We used control non-targeted siRNA (siCTRL, D-001810-01-20), and siRNA targeting STAT3 (siSTAT3) (M-003544-00-0005), STAT1 (siSTAT1) (M-003543-00-0005), IL6ST (siIL6ST) (M-003543-00-0005), SMAD2 (siSMAD2) (M-003561-01-0005), SMAD3 (siSMAD3) (M-020067-00-0005) and RELA (siRELA) (M-003533-00-0005) at the final concentration of 80 nM.

### 2.4. RNA Isolation, Reverse Transcription, and qPCR

An RNeasy Micro kit (Qiagen, Courtaboeuf, France) was used to extract total RNA [38]. A First-Strand cDNA kit (Thermo Fisher Scientific) was used to synthesize cDNA. RT-qPCR was performed using Power SYBR Green mix (Life Technologies), with a QuantStudio 3 analyzer (Thermo Fisher Scientific). Custom primers were designed with Primer-Blast online, and their efficiency and specificity were determined for each pair by RTqPCR on a serial dilution of cDNA samples. Relative quantification (2^−dCt^) was used to calculate the expression levels of each target gene, normalized to *CYCLOPHILIN-A* transcripts. The list of primers is presented in Appendix A.

### 2.5. Transcriptome Analysis

Total RNA was isolated using a RNeasy Micro kit (Qiagen, Hilden, Germany). The quality of the RNA was assessed by a bioanalyzer 2100 (Agilent, Santa Clara, CA, USA). In total, 100 ng of total RNA was used for each library. The RNA samples were processed with the TruSeq mRNA Standard kit (Illumina) according to manufacturer’s instructions. Libraries were sequenced with the NextSeq 500 on a High Output flow cell. In total, 38 ± 6 million read pairs were obtained for each sample. After sequencing, a primary analysis based on AOZAN software (ENS, Paris, France) [39] was applied to demultiplex and control the quality of the raw data (based on FastQC modules/version 0.11.5). Bioinformatic Alignment was performed with STAR Version 2.5.2b and Ensembl Homo GRCh38p12 release 93 as reference. Differential analysis was performed with R version 3.5.1 (2 July 2018). We used the standard DESeq2 normalization method (DESeq2′s median of ratios with the DESeq function), with pre-filtering of the reads and genes (reads uniquely mapped on the genome, or up to 10 different loci with a count adjustement, and genes with at least 10 reads in at least 3 different samples). Following the package recommendations, we used the Wald test with the contrast function and the Benjamini–Hochberg FDR control procedure to identify the differentially expressed genes. R scripts and parameters are available on GitHub (https://github.com/BSGenomique/genomic-rnaseq-pipeline/releases/tag/v1.0420), (accessed on 1 September 2021) RNAseq and raw data are available in the NCBI’s Gene Expression Omnibus (GEO) database (accession GSE184795).

The data were analyzed using several softwares, such as Biojupies created by the Ma’ayan laboratory and used to create the Heatmap and analysis pathways (https://amp.pharm.mssm.edu/biojupies/upload/table (accessed on 1 September 2021)) [40] and iDEP.91 established by Ge S.X et al. to confirm (http://bioinformatics.sdstate.edu/idep/ (accessed on 1 September 2021)) [41]. HeatMap illustrations of a set of genes selected on the RNAseq were performed on Morpheus (https://software.broadinstitute.org/morpheus/ (accessed on 1 September 2021)).

### 2.6. Immunoblotting

For immunoblot assays, the cells were lysed in RIPA buffer and sonicated. Equal amounts of protein (15 μg) were separated in 4–12% Bis-Tris gel (Thermo Fisher Scientific) and transferred onto a PVDF membrane using an iBLOT2 Dry Blotting System (Thermo Fisher Scientific). Membranes were blocked with 5% milk and immunoblotted with the following primary antibodies overnight at 4 °C: SOX9 (1/500; ab5535; Millipore), MAFA (1/500; Gift from A. Rezania, Betalogics), PDX1 (1/2000; [42]), ACTIN (1/2000; A5441, Sigma Aldrich), PSTAT3 (1/1.000; 9131, Cell signaling), STAT3 (1/1.000; 9139, Cell signaling), PSTAT1 (1/1000; 7649, Cell Signaling, Danvars, USA), STAT1 (1/1000; 9176, Cell signaling), PSMAD2/3 (1/1000; 8828, Cell signaling), RELA (1/250, sc8008, Santa Cruz Biotechnology). Species-specific HRP-linked secondary antibodies (Cell signaling) were used for detection after washing and visualization was performed on an ImageQuant LAS 4000 following ECL exposure (GE Healthcare, Velizy, France).

### 2.7. Glucose-Stimulated Insulin Secretion (GSIS)

EndoC-βH1 cells were seeded onto Matrigel/fibronectin-coated 12-well plates at 3.5 × 10^5^ cells/well. EndoC-βH1 cells were treated for 48 h with CM diluted by half in the absence or presence of TNFα (1.000 U/mL, R&D). Then, they were starved in DMEM (Thermo Fisher Scientific) containing 0.5 mM glucose for 24 h, washed twice and then preincubated in Krebs–Ringer bicarbonate Hepes buffer (KRBH) containing 0.2% BSA in the absence of glucose for 1 h. Insulin secretion was measured following a 60 min incubation with KRBH containing 0.2% BSA that contained varying glucose concentrations or KCl. For insulin content measurements, EndoC-βH1 cells were lysed in the culture wells in 50 mM Tris-HCl, pH 8.0, 150 mM sodium chloride, 1.0% Igepal CA-630 (NP-40), 0.5% sodium deoxycholate, 0.1% sodium dodecyl sulfate (Thermo Fisher Scientific), and anti-protease tablets (Roche) for 20 min on ice. Insulin secretion and content were measured by ELISA (Mercodia AB, Uppsala, Sweden).

### 2.8. Statistics

The graphs were constructed with Prism software (GraphPad). The results are presented as the mean ± SEM (Standard Error of the Mean). The number of experiments is indicated in the figure legends. Differences from control were evaluated using one-way ANOVA following by Dunnet’s multiple comparison test, or when the normality test was not passed a Kruskall–Wallis followed by Dunn’s multiple comparison test was used. For more than two groups, two-way ANOVA and a Bonferroni post-test was applied. A *p*-value less than 0.05 was considered significant.

## 3. Results

### 3.1. EndoC-βH1 Cells Are Sensitive to Conditioned Medium from Human Omental Stromal Cells

We first developed an assay to test the effects of soluble factors produced by adipose tissue-derived stromal cells on human beta cells (Figure 1a). We obtained human omental adipose tissue biopsies from morbidly obese candidates for bariatric surgery. We isolated and amplified stromal cells. They were cultured until confluence in the absence or presence of TNF-α to create a pro-inflammatory environment, or differentiated into adipocytes. Conditioned media (CM) were next prepared by incubating adipose tissue-derived cells for 48 h in EndoC-βH1 culture medium, and then filtered and frozen. Next, a human beta cell line EndoC-βH1 was cultured with omentum-derived CMs diluted by half for 48 h, and its effects on gene expression were quantified by RNA-sequencing. Around 1000 genes were differently expressed between the control conditions and treatment with CM by the stromal cells (1157 and 878 when stromal cells were pretreated or not with TNF-α respectively), and 141 genes were deregulated in EndoC-βH1 pretreated with CM from stromal cells differentiated into adipocytes. The CM from omental stromal cells activated inflammatory pathways (Figure 1b) and up-regulated the STAT pathways (Figure 1c). As examples, the expressions of STAT1, STAT3, IL6ST and IL13RA1 were induced upon CM treatment (Figure 1c). The SMAD pathway was also induced by CM treatment, with differential effects of stromal- vs. adipocyte-derived CM. Indeed, stromal-derived CM and stromal/TNFα-derived CM induced the expressions of SMAD7, SMURF2, SKIL and BMPR2, which was not the case for adipocyte-derived CM (Figure 1d). We also noted a sharp decrease in the expressions of many genes involved in lipogenesis and cholesterol synthesis upon treatment with CM from omental-derived cells (Figure 1b,e). Finally, many beta cell-specific genes were down-regulated upon CM treatment. This was, for example, the case for transcription factors such as *NKX6-3, NKX6-1, MAFA, PDX1* and *MNX1*. It was also the case for the RNA-encoding proteins implicated in insulin secretion, such as *CHGA, GCK, SCG5* and *SCGN,* or metabolic enzymes such as *AACS, ENO2,* and *HADH* (Figure 1f) [43,44,45,46,47,48]. Altogether, the data suggest that soluble factors derived from adipose stromal cells could regulate beta cell identity/fate. 

Based on RNA-seq analyses, we first investigated the impact of the STAT pathway on the regulation of beta cell reprogramming. Gene expression analyses revealed that omental-derived CM increased *STAT1* and *STAT3* (Figure 2a and Appendix A for quantification). Time course analyses via Western blotting indicated that the STAT1 and STAT3 levels increased after 24 h of treatment, while STAT1 and STAT3 phosphorylation was already observed after 1 h of treatment; STAT1 phosphorylation was transient, and STAT3 phosphorylation was stable with time (Figure 2a and Appendix A for quantification). We also noted that STAT targets such as *SOCS3*, *FBG*, *IL6ST* (Figure 2a), *IL13RA1, PARP9, JUNB, DTX3L, PLOD2* and *SBNO2* (Appendix A) were increased in EndoC-βH1 upon treatment with CM from omental-derived cells, indicating that the whole STAT pathway was activated by CM treatment. We next analyzed the SMAD pathway. We observed that treatment with omental-derived CM induced *SMAD7, BMP5* or *SKIL* mRNA level increases (Figure 2b). Moreover, this treatment (either 1 or 48 h) induced SMAD2/3 phosphorylation (Figure 2b and Appendix A for quantification). We also observed signs of induction of the NFKB pathway with increased *NFKBIA* and *IRF1* mRNA levels and RELA (p65) protein following CM treatment (Figure 2c).

We next validated, via RT-qPCR, the effects of CM derived from omental-derived cells on the expressions of genes involved in lipogenesis and cholesterol synthesis. Specifically, RT-qPCR indicated a decrease in SREBF1, SCD, FASN, PCSK9, LDLR and HMGCR upon treatment with CM from omental-derived cells (Figure 2d).

We next used RT-qPCR to analyze the effects of CM from omental-derived cells on beta cell-specific genes. We observed a sharp decrease in beta cell-enriched transcription factors such as *MAFA*, *NKX6.1* and *PDX1* (Figure 3a). We also observed a decrease in mRNA-encoding beta cell-specific hormones, such as *INS pre-mRNA* and *IAPP* (Figure 3a). Time course analyses indicated that the decrease in *NKX6-1* mRNA level was observed as early as after 6 h of treatment, while the effects on the other tested genes required at least 12 h (Figure 3b). CM from omental adipose tissue, cultivated as an explant, also showed a decrease in beta cell markers, but to a lesser extent (data not shown). Interestingly, CM from omental stromal cells induced increases in the expression levels of *HES1, MYC, PAX4* and *SOX9*, all markers of beta cell dedifferentiation [20,21,49] (Figure 3c). This observation was confirmed at the protein level with the induction of dedifferentiation markers, such as SOX9, and the reduction in beta cell markers, such as MAFA and PDX1 (Figure 3d).

CMs from omental stromal cells pre-treated with TNF-α were the most effective in modulating beta cell gene expression. In order to determine whether this was due to traces of remaining TNF-α in the CM, we looked at the effect of recombinant TNF-α on beta cell gene expression. The RT-qPCR analyses indicated that TNF-α had no additional effect on the expression of beta cell markers, such as *MAFA*, and dedifferentiation markers such as *HES1* (Appendix A). Then, we tested the effect of the CM from omental stromal cells on EndoC-βH1 cell survival and function. Neither the survival of EndoC-βH1 cells, nor their ability to secrete insulin in response to glucose, was affected following a 48 h CM treatment (Appendix A).

Hence, CM from omental stromal cells leads to several modifications in EndoC-βH1: activation of the STAT, SMAD and NFKB pathways, reductions in metabolic processes such as lipogenesis and cholesterol synthesis, and losses of beta cell identity.

### 3.2. CM from Subcutaneous Stromal Cells Does Not Recapitulate the Effects of CM from Omental Stromal Cells

We then compared the effects of omental vs. subcutaneous stromal cells on beta cell-specific gene expression (Figure 4a). As expected, subcutaneous stromal cells had a strong adipogenic potential (more adipocytes, higher expression of AP2), while omental cells had a pro-inflammatory profile based on the *IL6* and *TGFB1* mRNA expression levels (Figure 4b). Importantly, CM from omental stromal cells (either control, inflamed or differentiated into adipocytes) decreased the expressions of beta cell genes (*MAFA, NKX6-1, PDX1, INSULIN* and *IAPP*), while this effect was not observed with CM from subcutaneous stromal cells (Figure 4c). It is noteworthy that a decrease in *NKX6-1* and *PDX1* was observed with CM from subcutaneous stromal cells differentiated into adipocytes (Figure 4c). Experiments performed with CM from adipose tissue fractions from 26 patients revealed the robustness of the effect (Figure 4c). CM from subcutaneous stromal cells did not increase the expressions of dedifferentiation markers such as *HES1, MYC, PAX4* and *SOX9*, which differed from the induction observed with CM from omental stromal cells (Figure 4d).

Taken together, CM derived from human omental but not subcutaneous stromal cells decreased human beta cell identity. 

### 3.3. Implication of STAT1 and STAT3 Pathways in the Effects of CM from Omental Stromal Cells on Beta Cell Identity

We silenced STAT3 in EndoC-βH1 using siRNA (siSTAT3). The knock-down was efficient, as demonstrated by the sharp decrease in *STAT3* mRNA under basal and stimulated (IL6 treatment) conditions (Figure 5a,b), and the absence of P-STAT3 following IL6 treatment (Figure 5b and Appendix A for quantification). SiSTAT3 treatment also inhibited the induction of STAT3 by CM from omental stromal cells at the mRNA, protein and phosphorylation levels (Figure 5a,b). Knock-down also led to a decreased expression of many STAT3 canonic targets, such as *SOCS3, IL6ST* and *FGB* (Figure 5c), and additional ones such as *NFKBIZ* and *C2CD4A* [50,51,52,53] (Figure 5d). Interestingly, we observed that STAT3 knock-down reversed the induction of *HES1* observed upon CM treatment (Figure 5d). On the other hand, STAT3 silencing did not reverse the reduction in beta cell markers (*PDX1*, *MAFA*, *NKX6-1*) upon CM treatment (Figure 5e). Similar results were obtained by silencing STAT1 under the same experimental conditions (Appendix A). 

We finally tested the hypothesis that omental stromal cells secrete IL6-like cytokines that could act on beta cells through the STAT pathway. Then, we blocked IL6-like cytokines signaling using siRNA against IL6ST, one of the subunits of the IL6 receptor [54,55,56,57]. Thus, silencing was highly effective, based on the decrease in *IL6ST* mRNA under basal and stimulated conditions, and the altered STAT1 and STAT3 expressions (both at the mRNA and protein levels), as well as their phosphorylation under stimulated conditions (Figure 5f and Appendix A for quantification). Consequent to siIL6ST, we observed reduced expressions of STAT3 targets, such as *SOCS3* and *FGB* (Figure 5f). However, IL6ST knock-down did not reverse the decrease in beta cell markers (*PDX1*, *MAFA*, *NKX6-1*), but it did reverse the induction of *HES1* that was observed upon CM treatment (Figure 5g).

To sum up, omental stromal cells signal to beta cells through the STAT pathways by modulating the genes involved inflammatory pathways. However, STAT signaling did not regulate the gene network involved in the modulation of beta cell identity, with the exception of HES1.

### 3.4. Implication of SMAD and RELA Pathways in the Effects of CM from Omental Stromal Cells on Beta Cell Identity

We repeated the above-described strategy by silencing the SMAD pathway using siRNA targeting SMAD2 and SMAD3. RT-qPCR indicated that siRNAs were specific to their respective targets (Figure 6a). Western blot analyses showed that the CM from omental stromal cells induced SMAD2/3 phosphorylation (Figure 6b). This induction was specifically inhibited when SMAD2 was silenced, but was resistant to SMAD3 silencing (Figure 6b and Appendix A for quantification). Moreover, siSMAD2 (but not SMAD3) decreased the induction of the SMAD2/3 target gene, SMAD7 (Figure 6b) [58,59]. Finally, the effect of CM on beta cell markers (*MAFA*, *NKX6-1*, *PDX1*) was not reversed by either siSMAD2 or siSMAD3 (Figure 6c). Thus, omental stromal cells signal on beta cells through SMAD2; however, this signaling is not implicated in the regulation of beta cell identity. 

We finally silenced RELA, a subunit of NF-kB (also named p65). The knock-down was effective, with a sharp decrease in *RELA* mRNA and its target *NFKBIA* (Figure 6d). While this silencing had no effects on beta cell markers (*MAFA*, *NKX6-1*, *PDX1*) (Figure 6e), it reduced the induction of SOX9 observed upon CM treatment, at both the RNA and protein levels (Figure 6f and Appendix A for quantification).

Taken together, omental-derived factors signal to human pancreatic beta cells through different pathways (STAT, SMAD2 and RELA) to promote beta cell inflammation. Interestingly, the STAT and RELA pathways were restricted to the regulation of dedifferentiation markers, such as HES1 and SOX9, altering beta cell identity. 

## 4. Discussion

In this study, we demonstrate that soluble factors derived from stromal cells isolated from human adipose tissue signal on human beta cells and modulate their identity. 

We developed here a model to test the effects of a conditioned medium (CM) derived from adipose tissue on human beta cells. We produced CMs from a large series of either omental or subcutaneous human adipose tissue fat pads, and tested their effects on the human beta cell line EndoC-βH1 [18]. This cell line has been characterized as a valid model to study human beta cells, and can now be used for screenings to identify novel drug target candidates [60]. EndoC-βH1 cells were also validated as a model to study human beta cells under patho-physiological conditions, such as dedifferentiation [20], and sensitivity to lipotoxicity [61] and viral infection [21].

We observed that CMs from human omental stromal cells gives rise to an inflammatory signature in EndoC-βH1 cells. This was, for example, highlighted by the induction of the STAT pathway. The importance of this pathway in pancreatic beta cells has been described in the past in great detail. As examples, the STAT3 pathway has been characterized as a regulator of beta cell mass; it is implicated in the induction of endocrine cell differentiation from pancreatic ductal cells [32], and represents a modulator of beta cell cycling under specific conditions, such as pancreatic injury [33]. STATs have also been implicated as important players in the signaling of cytokines, such as IFN-γ [10] and IFNα [62], in pancreatic beta cells, and in the regulation of cytokines-induced apoptosis [34]. It was also found that IL6 signals through STATs in pancreatic beta cells [63]. Interestingly, our data indicate that CM from omental stromal cells induced STAT1 and STAT3, at both the mRNA and protein levels. CM also induced the phosphorylation of both STAT1 and STAT3. However, our time course experiments have indicated that while STAT1 phosphorylation was transient, peaking after 1 h of treatment, STAT3 phosphorylation was stable for up to 72 h. This temporal expression pattern is different from what has been observed in other contexts. As examples, the treatment of the rat insulinoma cell line INS1E with IL6 gives rise to the extremely transient phosphorylation of STAT3 [63], and while the treatment of EndoC-βH1 cells with IFNα induced both STAT1 and STAT3 phosphorylation, STAT3 phosphorylation is more transient than STAT1 phosphorylation [62]. We thus speculate that in our settings, STATs activation might not be due to a single factor derived from CM, but to a combination of signals. 

Our data indicate that CM from human omental stromal cells has a major effect on beta cell genes identity. We observed the decreased expression of beta cell-specific transcription factors such as MAFA, NKX6.1 and PDX1, and this was also the the case for their target genes INS and IAPP [36]. We observed a parallel increase in genes that are normally lowly expressed in, or absent from, mature beta cells such as HES1, MYC or SOX9. Altogether, this might suggest a process of dedifferentiation. In fact, beta cells are long-living factories that produce, store and secrete impressive amounts of insulin [64,65]. It has been observed that in a number of models that mimic type 2 diabetes, beta cells lose their mature phenotype and shift to an immature state named dedifferentiation [11,12]. This process of dedifferentiation was observed in many conditions, such as in chronic hyperglycemia [11], and also in mouse models of type 1 and type 2 diabetes [12,66]. We previously validated EndoC-βH1 cells as a model of human beta cell dedifferentiation following virus infection [21], growth factor stimulation [20], and lipotoxic stress [61]. Here, we have added CM from omental stromal cells to the list of conditions that induce human beta cell dedifferentiation. Intriguingly, this negative effect on beta cell identity was specific for omental CM, and was not replicated with CM from subcutaneous stromal cells. This major difference is not understood at this stage. However, it is important to keep in mind that stromal cells isolated from omental or subcutaneous adipose tissues differ at several levels. For example, subcutaneous stromal cells are more adipogenic, while visceral stromal cells are more pro-inflammatory [67,68]. The fact that CM from subcutaneous and omental cells differ in their capacity to induce the loss of beta cell identity might be used in differential screening aiming at identifying specific factors, or lipids and pathways, that induce beta cell dedifferentiation. Indeed, at this stage, we do not know the identity of such signals. Multiplex array and mass spectrometry will be used in the future to compare the secretome of omental versus subcutaneous adipose stromal cells, with the objective of identifying specific omental factors that modulate beta cell identity. Note that in the present study, we used stromal cells isolated from obese patients. As similar samples from lean controls were not available, we did not study their effects, which represents a limitation in our current study.

In the present study, we have employed a loss-of-function-based approach wherein we knocked-down specific genes. This siRNA-based approach is highly efficient, as demonstrated in this study and in previous ones [61]. However, none of the siRNA-mediated losses of function (STAT1, STAT3, IL6ST, SMAD2, SMAD3, RELA) reversed the decreased expression of beta cell-specific transcription factors observed upon treatment with CM from omental cells. However, we learned from such experiments that IL6 is not the factor in CM that induces beta cell dedifferentiation. Indeed, our data indicate that IL6 is efficiently expressed by omental in comparison to subcutaneous stromal cells and IL6 signals in EndoC-βH1 cells. However, by efficiently blocking the IL6 pathway using siRNA against IL6ST, we did not reverse dedifferentiation. We also learned that blocking one specific pathway was not sufficient to reverse the loss of identity induced by CM from omental stromal cells, which might suggest that the induction of more than one pathway is necessary to observe this loss of identity phenotype. Of note, the knock-down of either STAT3 or IL6ST reversed the induction of *HES1* observed upon CM treatment, while silencing RELA reduced the induction of *SOX9*. This suggests that reversal might be possible. In this context, it is interesting to keep in mind that different screens are currently being performed to discover ways to protect against dedifferentiation. In this context, it was found that treatments with inhibitors of the TGFβ pathway [69] or with mu-opioid receptor agonists [70] reverse beta cell dedifferentiation. Whether such types of molecules reverse dedifferentiation induced by CM from omental stromal cells remains to be tested.

## Figures and Tables

**Figure 1 cells-11-00924-f001:**
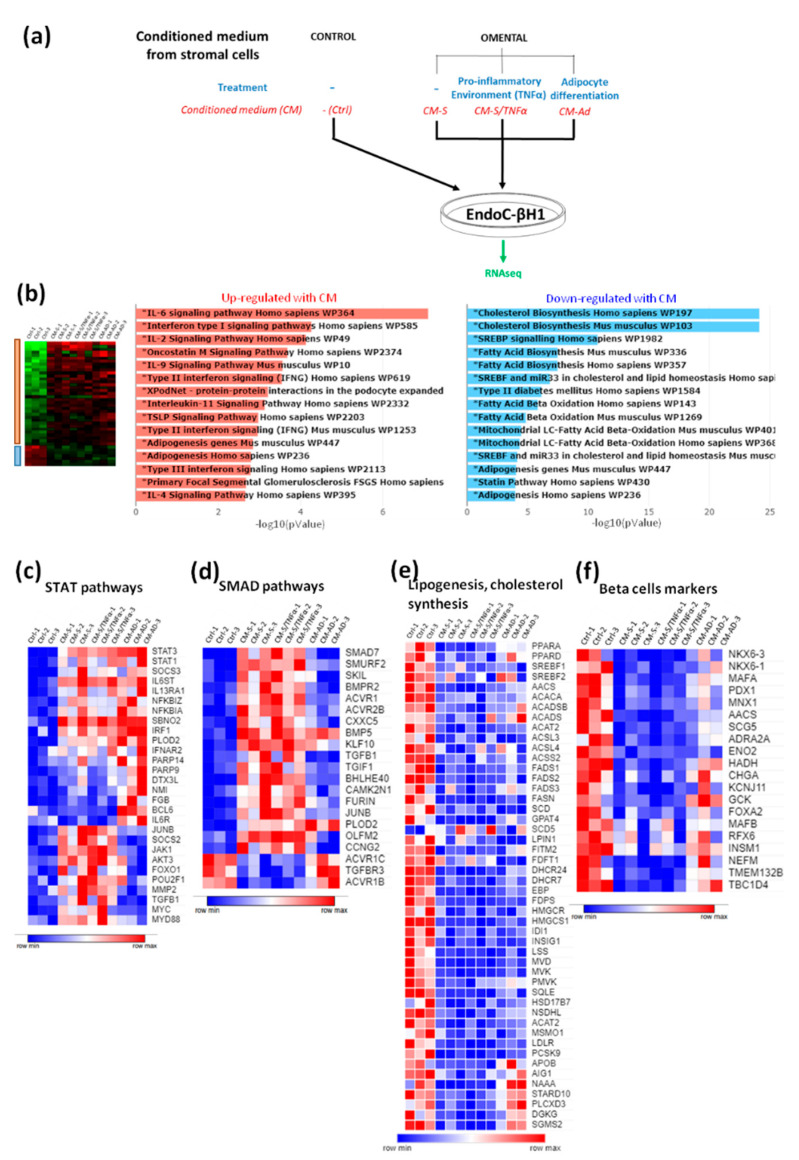
Impact of conditioned media (CM) from omental adipose stromal cells on EndoC-βH1 cells. (**a**) Experimental scheme: CM was prepared from stromal cells isolated from omental adipose tissue and cultured without (CM-S) or with TNF-α (CM-S/TNF-α), or differentiated into adipocytes (CM-AD). CMs were added to EndoC-βH1 cells and RNAs were prepared 48 h later for transcriptomic analyses. (**b**) Identification using Wikipathway of up- and down-regulated pathways in EndoC-βH1 cultured in omental conditioned media. (**c**–**f**) Heatmaps of genes of the STAT pathway (**c**), of the SMAD pathway (**d**), of lipogenesis and cholesterol synthesis (**e**) and enriched beta cells.

**Figure 2 cells-11-00924-f002:**
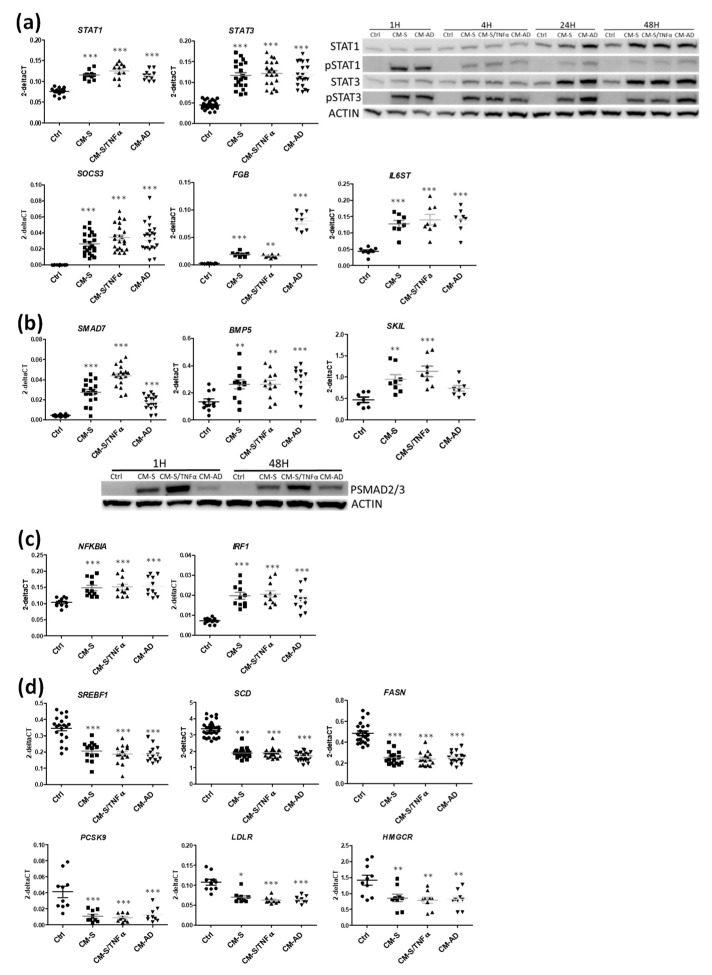
Effects of CM from omental adipose stromal cells on the STAT, SMAD and NFKB pathways and on lipogenesis and cholesterol synthesis genes in EndoC-βH1 cells. EndoC-βH1 cells were exposed to CMs. (**a**) Genes linked to the STAT pathway were analyzed by RT-qPCR after 48 h of treatment (n = 8–22). STAT1, pSTAT1, STAT3 and pSTAT3 were analyzed by Western blot after 1, 4, 24 and 48 h of treatment (n = 5). (**b**) Genes linked to the SMAD pathway were analyzed by RT-qPCR after 48 h of treatment (n = 8–18). PSMAD2/3 was analyzed by Western blot after 1 and 48 h of treatment (n = 6). (**c**,**d**) Genes linked to the NFKB pathway and to lipogenesis and cholesterol synthesis were analyzed by RT-qPCR after 48 h of treatment (n = 9–24). Data are represented as mean ± SD. * *p* < 0.05, ** *p* < 0.01, *** *p* < 0.001.

**Figure 3 cells-11-00924-f003:**
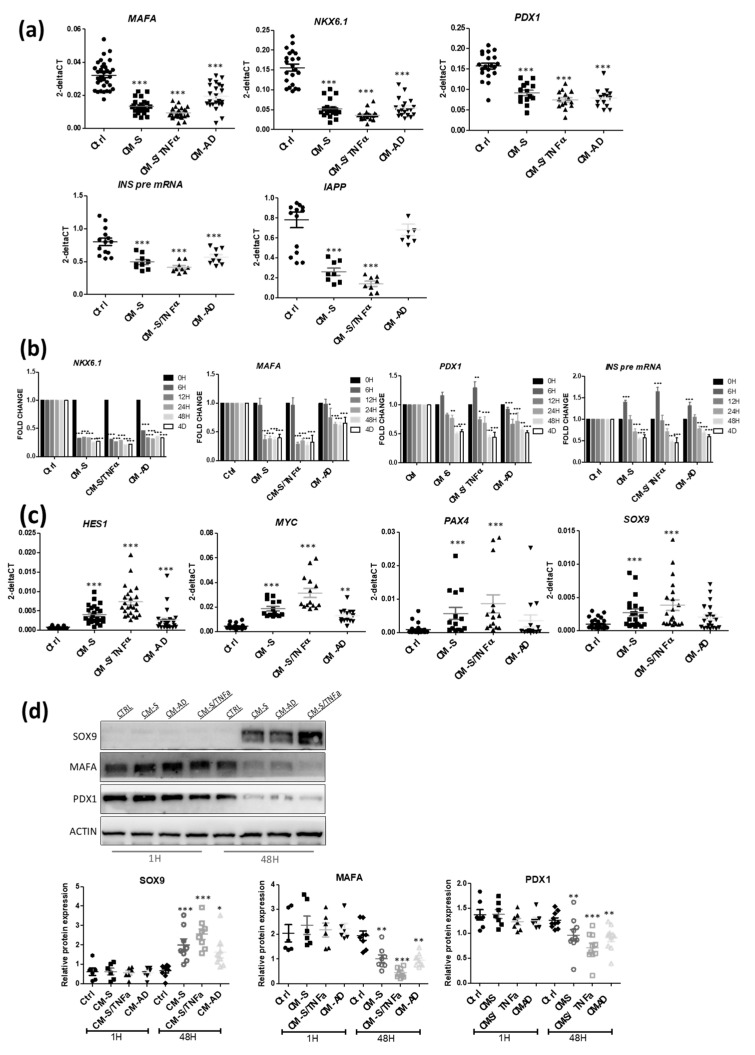
Effects of CM from omental adipose stromal cells on beta cell markers in EndoC-βH1 cells. (**a**) EndoC-βH1 cells were exposed to CMs from omental adipose stromal cells for 48 h. Beta cell-specific genes (*MAFA, NKX6-1, PDX1, INS pre-mRNA, IAPP*) were analyzed by RT-qPCR (n = 8–23). (**b**) Time course effect of CMs from omental adipose stromal cells on beta cell-specific genes (n = 3–5). (**c**) Effects of CMs from omental adipose stromal cells (48 h treatment) on beta cell dedifferentiation markers (*HES1, c-MYC, PAX4, SOX9*) (n = 14–23). (**d**) Western blot analyses of SOX9, MAFA and PDX1 following exposure to CMs for 1 and 48 h (n = 6). Data are represented as mean ± SD. * *p* < 0.05, ** *p* < 0.01, *** *p* < 0.001.

**Figure 4 cells-11-00924-f004:**
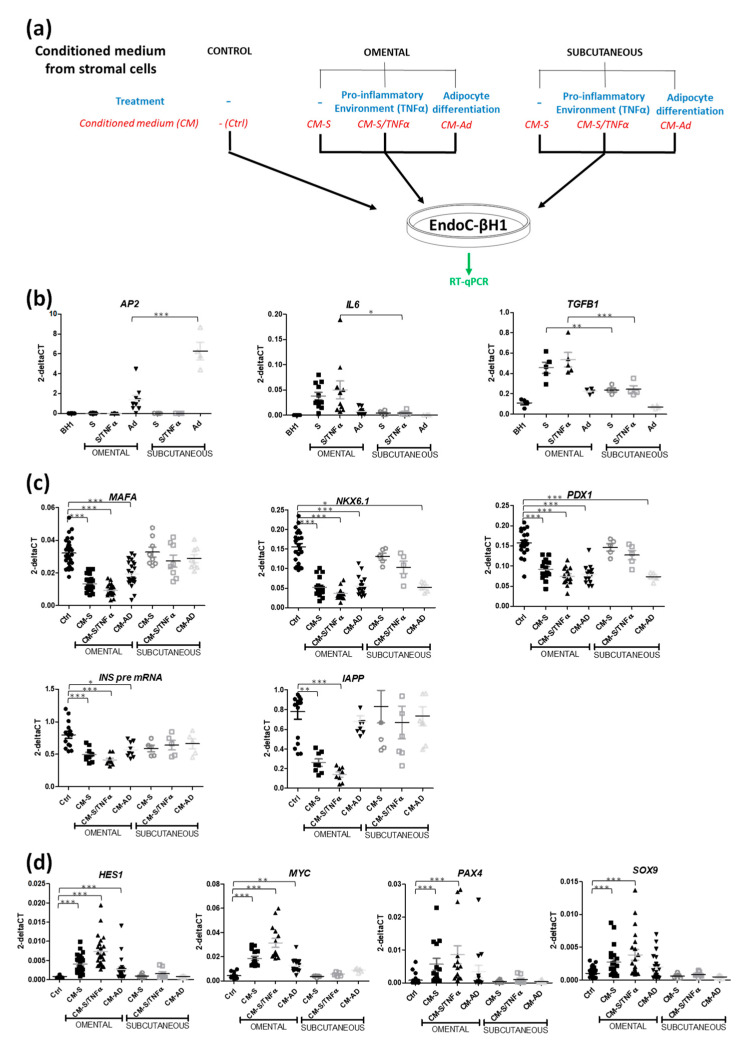
CM from subcutaneous stromal cells does not recapitulate the effects of CM from omental stromal cells. (**a**) Experimental scheme. (**b**) qPCR analyses of *AP2*, *IL6* and *TGFB1* mRNA in omental and subcutaneous cells (either control, inflamed or differentiated into adipocytes) (n = 4–10). (**c**,**d**) EndoC-βH1 cells were exposed to CMs from omental or subcutaneous stromal cells for 48 h. Beta cell-specific genes (*MAFA, NKX6-1, PDX1, INS pre-mRNA, IAPP*) (**c**) and dedifferentiation markers (*HES1, c-MYC, PAX4, SOX9)* (**d**) were analyzed by RT-qPCR (n = 5–23). Data are represented as mean ± SD. * *p* < 0.05, ** *p* < 0.01, *** *p* < 0.001.

**Figure 5 cells-11-00924-f005:**
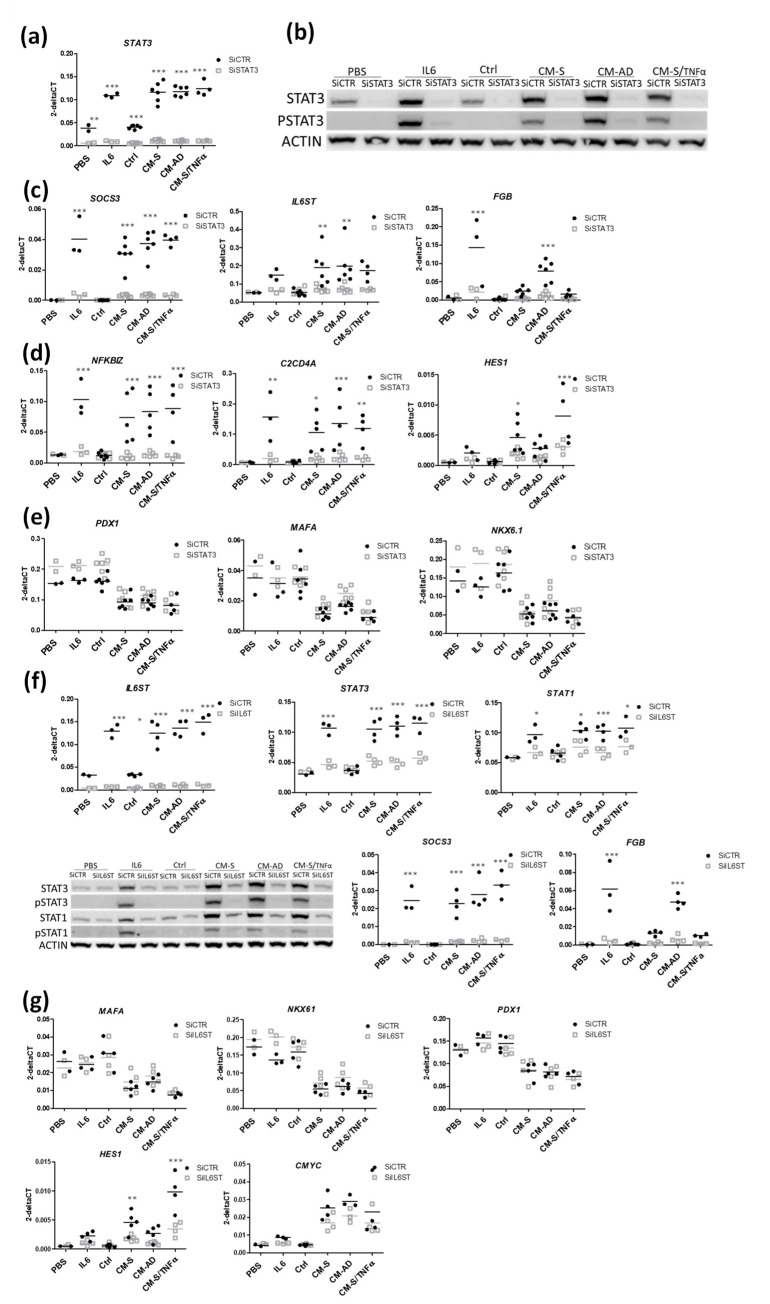
Role of STAT1 and STAT3 pathways in the effects of CM from omental stromal cells on beta cell identity. EndoC-βH1 cells were transfected with control non-target siRNA (siCTRL) or siRNA-targeting STAT3 (siSTAT3) (**a**–**e**) or IL6ST (siIL6ST) (**f**,**g**). Two days later, cells were treated with either IL6 or CM for 48 h. Analyses were performed by either RT-qPCR or Western blot. Data are represented as mean ± SD of 4–6 biological replicates. * *p* < 0.05, ** *p* < 0.01, *** *p* < 0.001.

**Figure 6 cells-11-00924-f006:**
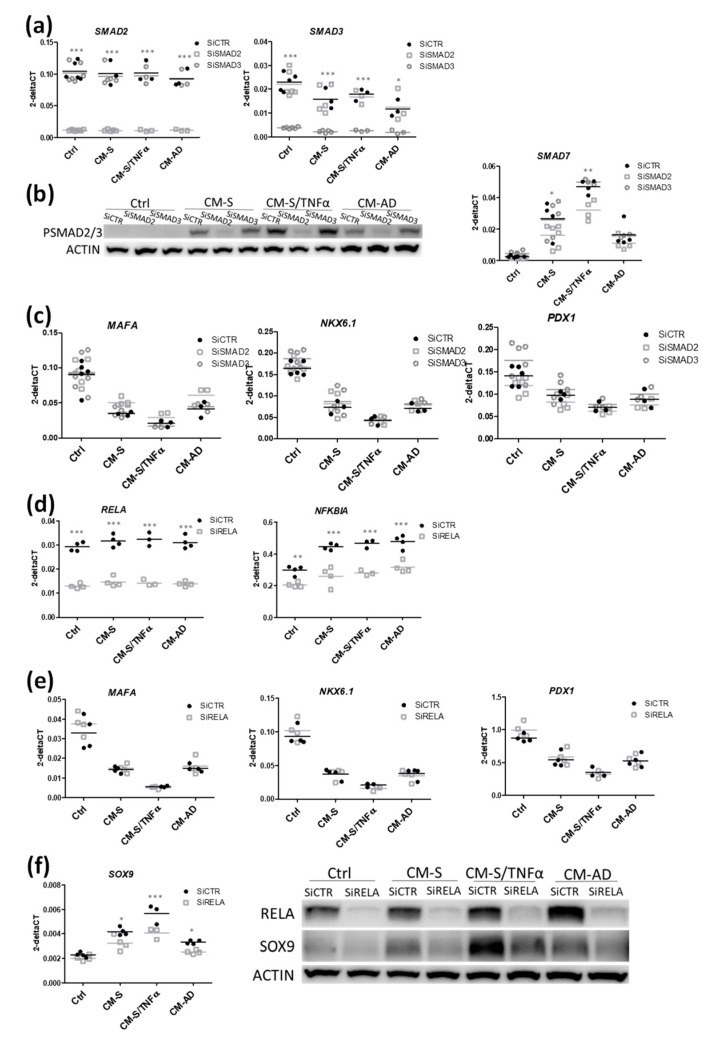
Role of SMAD and REL pathways in the effects of CM from omental stromal cells on beta cell identity. EndoC-βH1 cells were transfected with control nontarget siRNA (siCTRL), or siRNA targeting SMAD2 or SMAD3 (siSMAD2, siSMAD3) (**a**–**c**) or RELA (siRELA) (**d**–**f**). Two days later, cells were treated with either IL6 or CM for 48 h. Analyses were performed by either RT-qPCR or Western blot. Data are represented as mean ± SD of 4–12 biological replicates. * *p* < 0.05, ** *p* < 0.01, *** *p* < 0.001.

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
