# Peer review of "Loss of Human Beta Cell Identity in a Reconstructed Omental Stromal Cell Environment"

_cells, 2022, doi:10.3390/cells11060924_

Round 1

Reviewer 1 Report

The Authors have addressed all my requests.

Reviewer 2 Report

The authors have adequately answered my comments.

This manuscript is a resubmission of an earlier submission. The following is a list of the peer review reports and author responses from that submission.

Round 1

Reviewer 1 Report

This paper describes an extensive investigation of the impact of the conditioned medium (CM) of adipose tissue-derived stromal cells on gene expression by the human beta-cell line EndoC-ßH1. This cell line is a now well-validated model of human beta-cells and it is used here to determine how soluble factors produced by the omental or subcutaneous fat stromal fractions, treated or not with TNF-alpha or induced to differentiate in adipocytes, impact beta-cell dedifferentiation. The key findings are that the CMs from omental fat induce a decreased expression of major beta-cell transcription factors while inducing the expression of dedifferentiation markers. The effect of CM medium from subcutaneous fat is, in contrast, much less active in inducing these deregulations. Interestingly, the CMs of omental fat is shown to markedly increase the STAT, SMAD and RELA signaling pathways. However, silencing the expression of these signaling proteins does not prevent the negative effect of the CMs on induction of a dedifferentiated state in the EndoC.ßH1 cells.

It is known that adipose tissue depots, omental or subcutaneous, have different metabolic regulatory roles, and potentially negative impacts on beta-cell function during the progression towards type 2 diabetes. The present study, which is very well carried out, based on a large collection of human fat biopsies, and clearly presented, deconstructs the system of adipose tissue-beta-cell interactions and provides a strong basis for the future work to identify the diffusible molecules and beta-cell pathways involved in this cross-talk.

Minor comments:

One key control experiment is to test the effect of TNF-alpha directly on EndoC-ßH1. This has been performed but it is not indicated whether the same concentration of cytokine has been used to treat EndoC-ßH1 cells and the stromal fraction.

As glycolysis is a strong regulator of glucose-stimulated insulin secretion, and as beta-cell dedifferentiation is usually associated with deregulated expression of the key glycolytic enzymes, such as Gck or LDH, have the authors identified deregulated expression of glycolytic genes in the treated EndoC-ßH1? And what about the other key regulators of GSIS such the Abcc8 or Kcnj11?

Can the author elaborate on the significance of reduced lipogenesis and cholesterol biosynthesis in the treated EndoC-ßH1?

Figure 3: the labeling of the panels (b, c, …) does not match the text of the Results section.

Could the authors elaborate how they want to identify the adipose-derived signaling molecules? Perhaps to be shortly discussed in the Discussion?

Reviewer 2 Report

The current manuscript aims to study the paracrine effect of stromal cells isolated from subcutaneous and omental adipose tissue on human beta cells. The Authors found that conditioned medium derived from omental stromal cells stimulates several pathways such as STAT, SMAD and RELA in EndoC-βH1 cells. In addition, upon treatment, the expression of beta cell markers decreased while dedifferentiation markers increased. Finally, the Authors demonstrate that soluble factors derived from stromal cells isolated from human omental adipose tissue signal on human beta cells and modulate their identity, although They have not been able to identify specific molecules responsible for this cross talk.

The manuscript deals with a very interesting topic. However, further experiments are needed.

Comments:

A major limitation of the study is that adipose tissue derives only from obese patients. Omental stromal cells isolated from lean controls should be also analysed. In addition, the Authors should provide the clinical characteristics of patients: age, sex, BMI, waist, blood chemistry values (glucose, HbA1c%, cholesterol…), presence of co-morbidity, presence of diabetes or family history of diabetes, etc…

Before starting to list the results, the Authors should accurately describe (possibly by adding a figure) the pathways They analyze, also indicating the processes they regulate at the beta-cell level. This would make the manuscript more understandable.

I believe that the Authors should analyze whether the conditioned mediums are able to affect beta-cell survival and glucose-stimulated insulin secretion. This missing information is of primary importance.

In immunoblotting images of Fig. 2, the Authors should perform and show a quantitative analysis of the bands. In addition, it is not clear CM-s/TNFalpha is not used at 1h and 14h. Furthermore, the loading order should be improved: please show the time course of each condition (CM-s, CM-s/TNFalpha, CM-Ad) separately from others. Why not all the genes analyzed by qRT-PCR are then also analyzed at the protein level by immunoblotting?

In the Fig. 3d, PDX1 image is not representative: except condition CM-s/TNFalpha for 48h, all other bands show similar intensity. Please, use a more representative image.

For all immunoblotting images in figures 5 and 6, quantitative analyzes should be added.

Finally, a multiplex analysis should be performed in conditioned medium, in order to identify adipokines that could be responsible for the effects on beta-cells.